# Association of Ligamentum Flavum Hypertrophy with Adolescent Idiopathic Scoliosis Progression—Comparative Microarray Gene Expression Analysis

**DOI:** 10.3390/ijms23095038

**Published:** 2022-05-01

**Authors:** Shoji Seki, Mami Iwasaki, Hiroto Makino, Yasuhito Yahara, Miho Kondo, Katsuhiko Kamei, Hayato Futakawa, Makiko Nogami, Kenta Watanabe, Nguyen Tran Canh Tung, Tatsuro Hirokawa, Mamiko Tsuji, Yoshiharu Kawaguchi

**Affiliations:** 1Department of Orthopaedic Surgery, Faculty of Medicine, University of Toyama, Toyama 930-0194, Japan; hiroto@med.u-toyama.ac.jp (H.M.); k.kamei9786@gmail.com (K.K.); hayato_care@yahoo.co.jp (H.F.); makohna@hotmail.com (M.N.); kenta0419@yahoo.co.jp (K.W.); bstungbv103@gmail.com (N.T.C.T.); tatsuro.h13@gmail.com (T.H.); tsujimam@med.u-toyama.ac.jp (M.T.); zenji@med.u-toyama.ac.jp (Y.K.); 2Faculty of Engineering, University of Toyama, Toyama 930-8555, Japan; miwasaki@eng.u-toyama.ac.jp; 3Department of Molecular and Medical Pharmacology, Faculty of Medicine, University of Toyama, Toyama 930-0194, Japan; yasuhito.yahara@hotmail.co.jp; 4Department of Orthopaedic Surgery, Takaoka City Hospital, Toyama 933-8550, Japan; m.zaiki0309@gmail.com; 5Department of Trauma and Orthopaedic Surgery, Vietnam Military Medical University, Hanoi 100000, Vietnam

**Keywords:** ligamentum flavum, adolescent idiopathic scoliosis, microarray analysis, TGF-β signaling, elastic fiber, interleukin-6

## Abstract

The role of the ligamentum flavum (LF) in the pathogenesis of adolescent idiopathic scoliosis (AIS) is not well understood. Using magnetic resonance imaging (MRI), we investigated the degrees of LF hypertrophy in 18 patients without scoliosis and on the convex and concave sides of the apex of the curvature in 22 patients with AIS. Next, gene expression was compared among neutral vertebral LF and LF on the convex and concave sides of the apex of the curvature in patients with AIS. Histological and microarray analyses of the LF were compared among neutral vertebrae (control) and the LF on the apex of the curvatures. The mean area of LF in the without scoliosis, apical concave, and convex with scoliosis groups was 10.5, 13.5, and 20.3 mm^2^, respectively. There were significant differences among the three groups (*p* < 0.05). Histological analysis showed that the ratio of fibers (Collagen/Elastic) was significantly increased on the convex side compared to the concave side (*p* < 0.05). Microarray analysis showed that *ERC2* and *MAFB* showed significantly increased gene expression on the convex side compared with those of the concave side and the neutral vertebral LF cells. These genes were significantly associated with increased expression of collagen by LF cells (*p* < 0.05). LF hypertrophy was identified in scoliosis patients, and the convex side was significantly more hypertrophic than that of the concave side. *ERC2* and *MAFB* genes were associated with LF hypertrophy in patients with AIS. These phenomena are likely to be associated with the progression of scoliosis.

## 1. Introduction

Adolescent idiopathic scoliosis (AIS) is a three-dimensional spine curvature that progresses during the pubertal growth spurt [1]. The prevalence of AIS is approximately 0.5–4% of the population, and it affects girls more than boys [1]. Progression of the curvature and the associated deformity of the spine may lead to respiratory dysfunction [2] and back pain [3]. If the scoliotic curvature progresses > 25° in skeletally immature patients, treatment is warranted, starting with bracing. Surgical treatment is usually required at ≥40–50° scoliotic curvature to halt the progression of scoliosis with its associated problems of deterioration of self-image, deformation of the rib cage, and back pain [4]. Both treatments can be problematic; bracing can cause skin irritation, a temporary decrease in vital capacity, mild pain in the chest wall, and inferior rib deformation [5]. Although surgical treatment generally has an acceptable complication rate, scarring, neurological deterioration, major vascular injury, and deep wound infections can occur [6]. There are no prophylactic measures available for AIS. The establishment of a treatment for scoliosis based on its pathogenesis is expected to have enormous benefits for many patients in the future.

The pathogenesis of AIS is still not understood, although recent studies have reported the involved genes genome-wide association analysis in patients with AIS [7,8]. However, the understanding of the genetic variants related to the progression of AIS has not led to clinical relevance [9]. ScoliScore is a prognostic genetic test designed to evaluate the risk of curve progression in skeletally immature patients with AIS with Cobb angles of 10° to 25° [10]. There was not any association between the single nucleotide polymorphisms (SNP) used in ScoliScore and curve progression or curve occurrence in the French-Canadian population, although there are differences in race [11]. Clinical applications do not always arise from genetic associations. Other factors which may be related to the pathogenesis of AIS, such as leptin levels [12], spinal cord tethering [13], abnormal somatosensory evoked potentials [14], asymmetrical loading of the posterior parts of the vertebrae [15], osteopenia [16], melatonin-signal dysfunction [17], and insufficient serum vitamin D levels [18] have been reported. Considering the above, the pathogenesis of AIS has been considered to be multifactorial [19]. 

The ligamentum flavum (LF) is anatomically positioned in the posterior column of the spine, which covers the posterior aspect of the dura mater. LF hypertrophy is observed in aging [20], obesity [21], and mechanical stress [22] and can result in lumbar spinal stenosis [23]. Although dorsal shear force of the spine is one of the suspected causes of AIS [15], its mechanism is not clear. Shortening of the spinal canal is observed in scoliotic spines [24], leading to the hypothesis that posterior spinal elements tether and cause lordosis and rotation of the spine because the anterior parts of the vertebral bodies continue to grow [24]. Biomechanical analysis suggested that the synergistic effects of, not only the compressive force of the expanding anterior vertebral body resulting in a force driving the apical vertebral body out of midline, but also the tension forces on the posterior column resulting in a force keeping the spinal posterior elements in the normal position, play important roles in the progression of scoliosis [25]. 

There are few reports of the role played by the LF in patients with AIS. The purpose of this study was to identify the pathogenesis of AIS progression, focusing on the LF using microarray and gene profiling with the goal of elucidating the mechanism of scoliosis progression, which may point to new prognostic and treatment methods. 

## 2. Materials and Methods

### 2.1. Subjects

A total of 22 patients with AIS were enrolled in this study. These 22 patients with AIS (20 females, 2 males; mean patient age 14.2 years) underwent posterior spinal fusion and correction at our institution from 2017 to 2019 (Table 1). According to the classification of Lenke et al. [26], Lenke Type 1 and 2 patients numbered 16 and 6, respectively. The inclusion criteria were patients who had undergone AIS surgery and follow-up at our hospital. All patients were given a thoracic MRI for back pain or for checking neurological abnormality. Additionally, 18 adolescent patients who had no scoliosis and had been given a thoracic spine MRI for another reason were selected as controls (12 females, 6 males; mean patient age 14.8). The reasons for thoracic MRI were: 6 chronic back pain, 2 acute back pain, 3 scapular pain, 3 leg numbness, 1 leg motor weakness, 1 atrophy of leg muscle, 1 chronic hypochondrium pain, and 1 steppage gait. Prior approval for the study was obtained from the Ethical Review Board of the University of Toyama (consent no. I2013004 [25-4]). Informed consent to participate in this study and consent to instrumentation were obtained before surgery. 

### 2.2. MRI Imaging 

LF thickness of scoliosis was assessed at the facet joint level on an axial view of T2-weighted MRI as previously described [27,28]. The cross-sectional area of LF was a more sensitive measurement than the thickness of LF [27]. In order to analyze as comprehensively as possible, we measured the thickening of the LF using the cross-sectional area by MRI. The 18 control subjects were selected from the same age group who had taken thoracic spinal MRI images for other diseases such as back pain and had no scoliotic curvature. All control patients were measured at the cross-sectional area of LF in axial view between T7/8 and T9/10. Additionally, 22 scoliosis patients were evaluated by MRI for surgery on a Lenke type 1 or type 2 thoracic curve, as shown in Table 1. The level of measurement was decided by the apex of the main thoracic curvature, and the cross-sectional area of LF on both the concave and convex sides was measured and analyzed separately. The measurement was performed using commercial software (Synapse VINCENT, Fujifilm, Tokyo, Japan). The value was measured three times for each patient by one experienced spine surgeon, and the mean value was considered the LF thickness (Figure 1). 

### 2.3. Isolation of Human LF cells from AIS Patients

LF tissues were obtained from scoliotic patients who had undergone surgery at Toyama University Hospital. This study was approved by the Ethics Review Committee of our institution (approval no.: I2013004). Written informed consent was obtained from all patients before the collection of specimens. LF was cut into small fragments and submitted to sequential enzymatic digestion with collagenase. The cells were then filtered through a nylon mesh with a pore diameter of 70 μm (Corning Gilbert, Glendale, AZ, USA), centrifuged at 1600 rpm, and the supernatant was discarded. The resulting pellet was washed 2 times with phosphate-buffered saline. Cells were cultured in a 10 cm culture dish with Dulbecco’s Modified Eagle Medium under consistent culture conditions (37 °C, 5% CO_2_). Medium was supplemented with 10% Fetal Bovine Serum and 1% of an antibiotic mixture (100 U/mL penicillin, 100 mg/mL streptomycin). Medium was replaced every week. For subcultivation, the cells were detached with trypsin/ethylenediaminetetraacetic acid and expanded.

### 2.4. qPCR

Total RNA was prepared from LF cells or tissues using Isogen (Nippon Gene, Tokyo, Japan). Total RNA extraction and cDNA synthesis was carried out (PureLink RNA Mini Kit, Invitrogen, Waltham, MA, USA) and High-Capacity RNA-to-cDNA Kit (Invitrogen, MA, USA) by using a Thermal cycler, respectively. Extracted RNA was measured by Nano Drop One Spectrophotometer (Invitrogen, Waltham, MA, USA). Gene expression analysis was carried out (iTaq Universal SYBR Green Supermix, Bio-Rad, Hercules, CA, USA) based qPCR reaction on the CFX Connect (Bio-Rad, Hercules, CA, USA). All procedures were performed according to each manufacturer’s protocol. Data were normalized to those of GAPDH mRNA, and relative gene expression of LF tissues and cells was determined using ∆∆Ct method.

### 2.5. Hematoxylin-Eosin, Azan, Elastica Van Gieson Staining, and Immunohistochemistry 

LF tissues were fixed in 10% formalin and embedded in paraffin. Tissue sections (thickness: 4 μm) mounted on glass microscope slides were stained with HE, Azan, or EVG. Deparaffinized sections were stained with hematoxylin-eosin, Eosin Y solution (Wako, Osaka, Japan), to analyze LF tissue. After dewaxing, the sections were stained using hematoxylin and eosin (H and E) staining. For Azan staining, slides were incubated in prewarmed azocarmine G (40012, Muto Pure Chemicals, Tokyo, Japan) for 2 h, washed with distilled water, differentiated with aniline alcohol (40021, Muto Pure Chemicals Co., Ltd., Tokyo, Japan), briefly washed with acetic alcohol and distilled water, incubated in 5% phosphotungstic acid (40041, Muto Pure Chemicals Co., Ltd., Tokyo, Japan) for 2 h, briefly washed with distilled water, incubated in aniline blue/orange G solution (40052, Muto Pure Chemicals Co., Ltd., Tokyo, Japan) for 1 h, and then differentiated using 100% alcohol. For EVG staining, slides were stained with Weigert’s resorcin Fuchsin solution (233-01655, Wako, Osaka, Japan) for 1 h, washed with 100% alcohol, the nuclei stained with Weigert’s iron hematoxylin solution (298-21741, Wako, Osaka, Japan) for 10 min, washed with water, and then stained with Van Gieson solution F (221-01415, Wako, Osaka, Japan) for 3 min, and washed with 70% alcohol. After staining, slides were dehydrated, cleared, and mounted with a mounting agent for microscopy. We performed the method used to quantify the ratio of collagen fibers to elastic fibers as previously described [29]. To calculate the area of collagen and elastic fibers after EVG staining, we used the ImageJ software program (National Institutes of Health, Bethesda, MD, USA). As shown in Appendix A, in the case of elastic fiber, after converting EVG staining to 8-bit grayscale, the area of the gray-stained part was calculated using ImageJ. In the case of collagen fiber, the area of the grayscale image was calculated by ImageJ after black-and-white inversion. A total of three EVG stains with a cross-section perpendicular to the elastic fiber were selected and measured three times to calculate the average value.

### 2.6. Immunohistochemistry

For immunohistochemical analysis, after deparaffinization and rehydration, sections were digested with 500 U/mL testicular hyaluronidase for 30 min at 37 °C. Endogenous peroxidase activity was inhibited with 3% H_2_O_2_ containing 0.1% sodium azide for 5 min at room temperature. Sections were incubated overnight at 4 °C in a humidity chamber with rabbit polyclonal antibodies to the ERC2 (ab27250, Abcam, Cambridge, UK, 1:500) or rabbit monoclonal antibodies to MAFB (#41019, Cell Signalling Technologies, Danvers, MA, USA, 1:1000). Immunostaining reactions were performed using an Envision™+ Kit (K4007, Agilent Technologies, Santa Clara, CA, USA), followed by staining using 3,3′-diaminobenzidine (DAB) reagent.

### 2.7. Western Blotting 

We prepared cell lysates using a buffer (RIPA Lysis and Extraction Buffer 89900, Thermo Fisher Scientific, Waltham, MA, USA). Proteins (30 μg) were subjected to sodium dodecyl sulphate-polyacrylamide gel electrophoresis, transferred to a polyvinylidene fluoride membrane that was blocked with 5% skim milk, and then incubated with an anti-ERC2 antibody (ab27250, Abcam, 1:500), anti-MAFB antibody (#41019, Cell Signalling Technologies, 1:1000) an anti-ACTB antibody (#4970, Cell Signalling Technologies 1:1000) in 5% skim milk. An anti-rabbit IgG conjugated to horseradish peroxidase (#7074, Cell Signalling Technologies, 1:1000) was used as the secondary antibody in Tween 20/100 mM NaCl/10mM Tris-HCL (1:1000 dilution).

### 2.8. Microarray Analysis

Total RNA was extracted from the cells using a kit (PureLink RNA Mini Kit, Invitrogen). After RNA was qualified (Agilent 2100 Bioanalyzer), Cy3-labeled cRNA was synthesized from 50 ng of total RNA using a kit (Low Input Quick-Amp Labeling Kit, One-color, Agilent Technologies) and purified using another kit (RNeasy Mini Kit (Qiagen, Hilden, Germany). The concentration of amplified cRNA and dye incorporation was quantified using spectrophotometry (NanoDropOne spectrophotometer, Thermo Fischer) and hybridized using a kit (SurePrint G3 Human Gene Expression v3 8 × 60K Microarray Kit Design ID:072363, Agilent Technologies). After hybridization, arrays were washed consecutively (Gene Expression Wash Pack, Agilent Technologies). Fluorescence images of the hybridized arrays were scanned (SureScan Microarray Scanner, Agilent Technologies), and the scanned data were extracted with commercial software (Feature Extraction software ver. 12.1.1.1, Agilent Technologies). The raw microarray data are deposited in the National Center for Biotechnology Information Gene Expression Omnibus (GEO Series GSE85226). Gene expression analysis was performed (GeneSpring GX 14.9.1, Agilent Technologies). Each measurement was divided by the 75th percentile of all measurements in that sample at per chip normalization. The genes filtered by flags were detected in all samples and were subjected to further analyses.

### 2.9. Statistical Analysis

Data were compared by Mann–Whitney U-test, Pearson’s correlation coefficient, and Student’s t-tests were used for the statistical analysis using commercial software (JMP^®^ version 9, SAS Institute Inc., Cary, NC, USA). *p*-values < 0.05 were considered statistically significant.

## 3. Results

We first measured the thicknesses of the LF in the apex of thoracic AIS curvatures in the convex and concave side with MRI in patients and compared them with LF in non-scoliotic patients. Next, transcriptional profiling, microarray, Western blotting, and histological analyses of LF were performed.

### 3.1. Measurements of LF Hypertrophy in Patients with AIS

We first measured the cross-sectional area of the LF at the apex of the main thoracic curvature on the convex and concave aspects in Lenke type 1 or 2 patients with AIS, as shown in Figure 1a,b [27,28]. Figure 1c shows the surgical specimens of convex and concave apical LFs from a Ponte osteotomy [30]. The LF of the convex side was more hypertrophic than the LF of the concave side. There was also a significant difference (*p* < 0.05) between the non-scoliotic group and scoliotic LF thicknesses, as shown in Figure 1d. Next, we investigated whether convex LF hypertrophy occurred in periapical areas. Figure 1e shows that the LF thickness of apex−1 (cranial to the apex), apex, and apex+1 (caudal to the apex) was significantly more hypertrophic than that of the concave side but was not as thick as the convex side at the apex of the curvature (*p* < 0.05). 

### 3.2. Histological Analysis of LF Hypertrophy with Comparison between the Concave and Convex Sides

Next, we investigated the cellular properties of LF hypertrophy on the convex side occurring in periapical regions. Hematoxylin-eosin (HE), Azan, and Elastica van Gieson (EVG) staining were performed on LFs from both sides (Figure 2). LF of the convex side appeared to have a slightly denser fibrous structure than that of the concave side in HE staining (Figure 2a). Azan staining showed many areas that were strongly blue-stained on the convex side, suggesting an increase in collagen fibers (Figure 2b). EVG stain showed that the elastic fibers were sparse on the convex side because it was more disturbed on the convex than on the concave side (Figure 2b). According to an analysis of EVG stain, the ratio of fibers (Collagen/Elastic) was significantly increased on the convex side compared to the concave side (*p* < 0.01), indicating that collagen fiber density was increased and elastic fibers were not changed much on the convex side. 

### 3.3. Quantitative PCR (qPCR) Analysis of LF

Next, we performed gene expression analysis of LF hypertrophy on the convex and concave sides occurring in the periapical regions. Several types of collagen (type I, II, III, and X), cytokines such as transforming growth factor (*TGF*)-beta1, Interleukin (*IL*)-1, *IL*-6, tissue necrosis factor (*TNF*)-alfa, matrix metalloproteases (*MMPs*) such as *MMP2*, and growth factors such as connective tissue growth factor (*CTGF*) and platelet-derived growth factor (*PDGF*) have been reported as factors involved in LF hypertrophy [28,29]. LF tissues of neutral vertebrae (Control), periapical concave, and convex side were removed to release intervertebral space. The expression analysis of these tissues was performed with quantitative PCR (qPCR). Extracellular matrix (ECM) protein of the periapical concave and convex sides was significantly increased compared to that of control, as shown in Figure 3 (*p* < 0.05). Furthermore, the convex side of ECM protein was significantly increased compared to the concave side (*p* < 0.05). *IL-6*, *TGF*-beta1 and *CCN1* showed same expression pattern (*p* < 0.05).

### 3.4. Gene Expression Profiling of LF between Concave and Convex Side

Next, we compared gene expression profiling between the concave and convex sides of the apical region using microarray analysis of these tissues. Heatmap analysis (Figure 4a) showed relatively close gene profiling between the concave and convex aspects. Scatterplot analysis (Figure 4b) found 11 genes with different expression levels. Next, we used Western blots to confirm whether these genes were really different in expression level, even at the protein level. This revealed that the expression level of the protein in ERC2 (ELKS/RAB6-Interacting/CAST Family Member 2) was clearly different between the concave and convex sides (Figure 5a). Immunohistochemical analysis showed the ERC2 protein was abundant on the convex side, localized in the cytoplasm (Figure 5b). We further investigated why the increased expression of ERC2 was associated with LF hypertrophy on the convex side. The LF cells from controls were isolated and used with experimental study. We generated overexpression vectors of *ERC2*, and transfected them into isolated LF cells. *COL1A2, COL2A1, COL3A1, IL-6,* and *TGF-beta1* were significantly increased in the presence of elevated *ERC2* in LF cells. (Figure 5c). Overexpression of *ERC2,* as well as of *COL1A2, COL2A1*, *COL3A1*, *IL-6,* and *TGF-beta1* resulted (Figure 5c), suggesting that overexpression of *ERC2* may lead to LF hypertrophy on the convex side of the curvature.

### 3.5. Gene Expression Profiling of LF between Controls and the Convex Side of the Scoliotic Curvature

Next, we compared gene expression profiling between controls and the convex side. Microarray analysis with a heatmap (Figure 6a) showed relatively close gene profiles. Scatterplot analysis (Figure 6b) revealed 47 genes with significantly different expression levels. Western blots revealed that the expression levels of the gene product of V-maf musculoaponeurotic fibrosarcoma oncogene homolog B (*MAFB*) were clearly different among control, concave, and convex side LF tissues (Figure 7a), with the highest expression by the tissues on the convex side. Immunohistochemical analysis showed that *MAFB* appeared to be expressed in a cell population clustered around the fissures of the LF (Figure 7b). MAFB protein was abundant in the cytoplasm, especially in larger cells (Figure 7b). To investigate why increased expression of *MAFB* causes hypertrophy, we generated overexpression vectors of *MAFB* and transfected them into isolated LF cells. This resulted in increased expression of *IL-6* (Figure 7c), while *TGF- beta1* expression was unaffected. To further investigate the mechanism of hypertrophy on the convex side of the curvature, we performed an expression profile of the collagens by the simultaneous addition of *IL-6* and *TGF-beta1*. By adding these cytokines to control LF cells, *COL1A2* and *COL3A1* were significantly increased (Figure 7d). These phenomena suggest that *TGF-beta1* and *IL-6* may also lead to the hypertrophy of LF on the convex side of the curvature in patients with AIS.

## 4. Discussion

Although LF hypertrophy is seen in aging, we found a significant hypertrophic change in LF around the scoliotic curvature in patients with AIS. There were significant differences in LF hypertrophy between the concave and convex sides of the curvature. These phenomena suggest that the convex side of the curvature is exposed to stronger mechanical stress compared to that of the concave side. It has been reported that mechanical stress was considered a cause of LF hypertrophy based on computed tomographic, histological, and collagen content analyses of 161 lumbar LFs [22]. There has also been a report on the differences in mechanical stress between the convex and concave sides (6593.02 vs. 5022.48) in finite element analysis of an animal scoliotic model, with the mechanical stress on the convex side increased (31.27%) in the vertical direction of the vertebral body [31]. Another study found a difference in mean compressive stresses between the concave and convex sides of the scoliotic curves ranging between 0.1 and 0.2 MPa and showed that tension force was the strongest in the posterior aspect of the convex side [32]. The tension force in the posterior convex side may be related to the hypertrophic changes in the LF here. 

According to pathological analysis of LF hypertrophy in elderly patients, there is a loss of elastic fibers and an increase in collagenous fibers [33]. Similarly, in an animal model of LF hypertrophy, a decrease in elastic fibers and an increase in cartilaginous tissue and collagen fiber were observed [29,34]. It has been reported that the LF hypertrophic change was associated with the increased expression of TGF-beta1 [20,29]. In addition, *IL-6* mRNA and protein expression was increased in the hypertrophied LF in patients with lumbar spinal stenosis [35,36]. Other factors possibly involved in hypertrophic change of LF tissue include CTGF [29], PDGF [29], MMP2 [36], CCN5 [37], VEGF [38], and TIMP2 [39]. These cytokines are thought to be related to the TGF-beta pathway, LF tissue degradation, or its inhibition [40]. Our data showed a significant decrease in elastic fibers and an increase in collagenous tissue with EVG and Azan stain. The mRNA expression analysis of qPCR in LF tissue showed significantly increasing *TGF-beta1*, *IL-6,* and *CCN1* in LF of the convex side compared with the control and concave side. It was reported that *CCN1* was expressed by *TGF-beta**1* and enhanced *TGF-beta1/SMAD3*-dependent profibrotic signaling in fibroblasts [41]. These phenomena suggested that these cytokines may be associated with inducing a hypertrophic change of LF. 

Next, we investigated the gene expression profiling between the concave and convex sides and between the control and convex side because we would like to know more about the factors associated with the hypertrophic changes of LF. We identified two genes associated with the LF hypertrophy, *ERC2* and *MAFB*, by microarray analysis and Western blotting. The *ERC2* gene is highly expressed in the nervous system and is also widely expressed in skeletal muscle and ovaries [42,43]. Members of this protein family form part of the cytomatrix at the presynaptic active zone complex and function as regulators of neurotransmitter release [42,43]. However, there have not been any reports associating LF and *ERC2* as far as we investigated. At the same time, there have not been any reports of overexpression of *ERC2* inducing *TGF-beta1* expression or these collagens. This may be one of the other features of *ERC2* that have not been reported so far. *MAFB*, which is widely expressed in pancreatic α cells, renal podocytes, epidermal keratinocytes, hair follicles, and hematopoietic stem cells, also functions in embryonic urethral formation [44]. The protein encoded by this gene is a basic leucine zipper transcription factor that plays an important role in the regulation of lineage-specific hematopoiesis [44]. Additionally, MAFB protein is expressed in the tissue of Dupuytren’s cord [45], and *MAFB* and *Sox9* form a positive feedback loop that maintains cell stemness and tumor growth in vitro and in vivo [46]. *MAFB* expression is induced by *IL-10, IL-4*, and *IL-13* [44,47]. Our data showed overexpressed *MAFB* in LF cells highly induced *IL-6*. To the best of our knowledge, there have not been any reports of *MAFB* directly inducing *IL-6*. From our experiments, overexpression of *MAFB* did not directly increase the expression of collagens in LF cells (data not shown). However, increased expression of collagens was observed with the simultaneous addition of *TGF-beta1* and *IL-6*. These facts suggest that *MAFB* may be involved in the ligamentum flavum hypertrophy through elevated *IL-6* expression. Considering the progression of scoliosis (Figure 8), firstly, the LF hypertrophy may further progress through the expression of cytokines such as IL-6 and TGF-beta1 due to the increased expression of *ERC2* and *MAFB* in response to mechanical stress. Scoliosis may progress because hypertrophy of the LF, which is the supporting tissue of the posterior spine, induces an imbalance of the growth between the vertebral body and the posterior column, including pedicles, facets, and spinous processes. AIS likely progresses, inducing the overgrowth of the anterior vertebral body by keeping tethering posterior spinal elements due to LF hypertrophy. 

## 5. Conclusions

*ERC2* and *MAFB* genes were associated with LF hypertrophy through increasing *TGF-beta1* and *IL-6* in patients with AIS. 

## Figures and Tables

**Figure 1 ijms-23-05038-f001:**
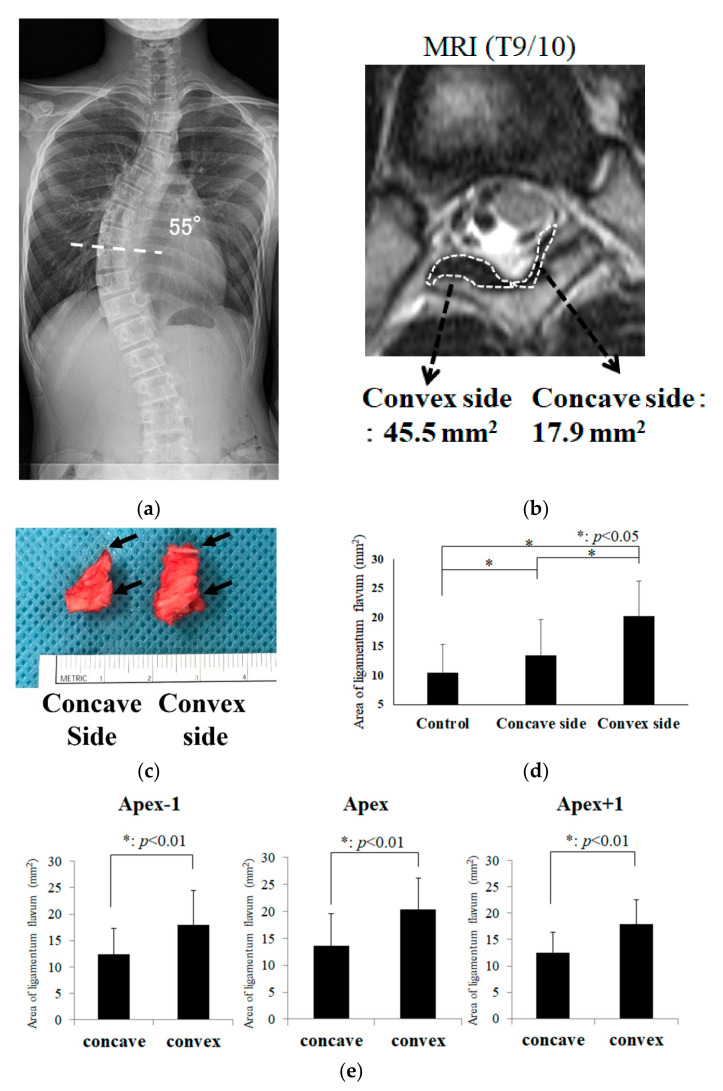
Measurement of LF thickness with MRI. (**a**) A 13-year-old female who had a 55° main thoracic curvature; (**b**) The cross-sectional area of LF was measured on the axial view of thoracic MRI. The axial spinal level was T9/10 as shown in (**a**); (**c**) The surgically extracted apical LF at T9/10 level. Arrows indicate the bone attachment sites (enthesis); (**d**) The thickness of LF was compared among 3 groups (18 subjects in the non-scoliosis [control] group, 22 subjects in concave side and convex side in apical main thoracic curvature); (**e**) Analysis of the thickness of periapical LF. Apex−1 indicates intervertebral space 1 vertebra cranial to the apex of the curvature.

**Figure 2 ijms-23-05038-f002:**
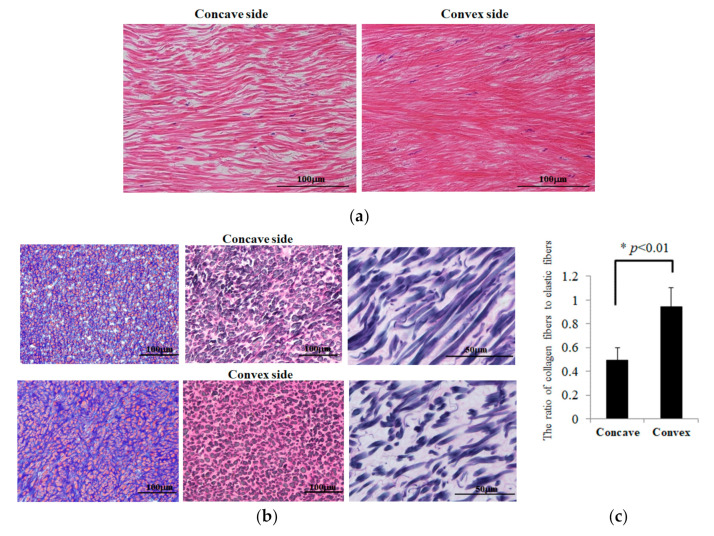
Hematoxylin-eosin (HE), Azan, and Elastica van Gieson (EVG) staining: (**a**) HE stain. (**b**) Azan (left side) and EVG (center and right side) stain. (**c**) Figure shows the ratio of collagen fibers to elastic fibers (Collagen/Elastic). * indicates *p* < 0.01.

**Figure 3 ijms-23-05038-f003:**
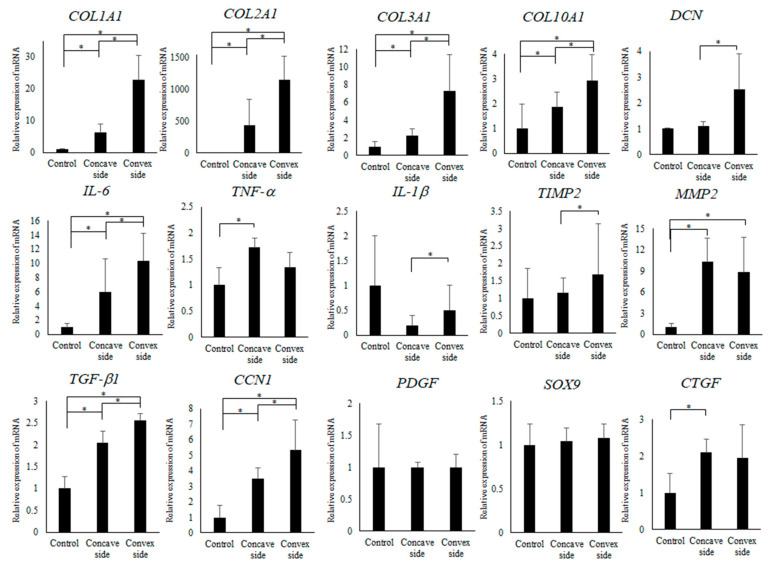
Quantitative PCR (qPCR) analysis of LF tissues in patients with AIS. Control indicates expression in LF tissue of intervertebral space in neutral vertebra. * indicates *p* < 0.05.

**Figure 4 ijms-23-05038-f004:**
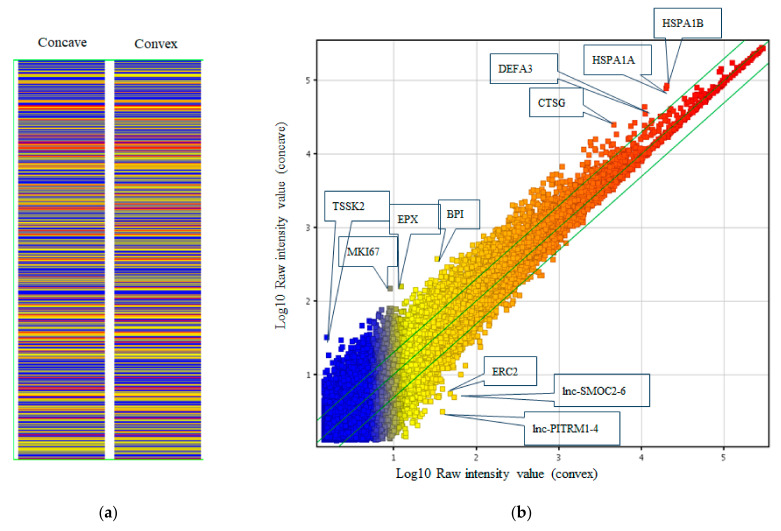
Microarray analysis of LF tissue comparing the concave and convex sides of the curvature. (**a**) Heatmap. (**b**) Scatterplot analysis.

**Figure 5 ijms-23-05038-f005:**
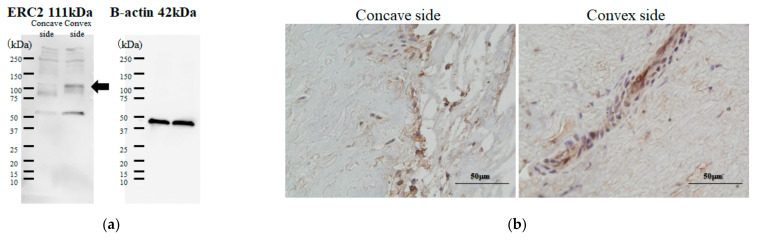
Expression and functional analysis of (ELKS/RAB6-Interacting/CAST Family Member 2) (ERC2) (**a**) Western blotting; (**b**) Immunohistochemical analysis of concave (left) and convex (right) side; (**c**) Expression analysis of collagens and cytokine under overexpressed *ERC2* in control LF cells.

**Figure 6 ijms-23-05038-f006:**
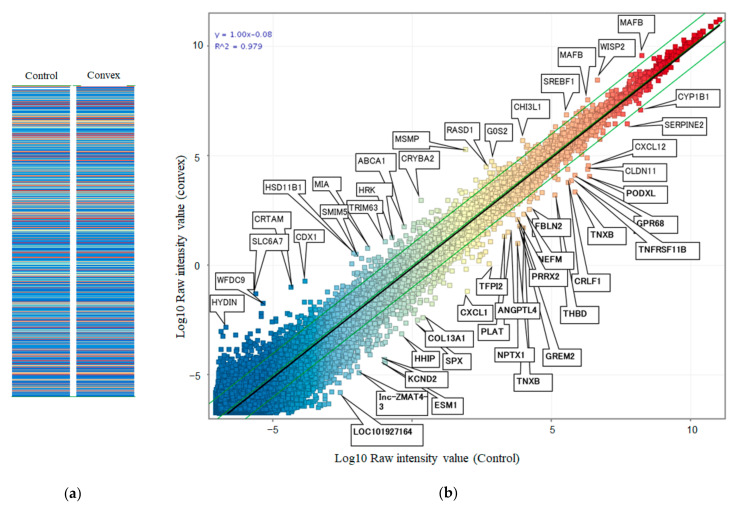
Microarray analysis of LF tissue between control and convex side. (**a**) Heatmap. (**b**) Scatterplot analysis.

**Figure 7 ijms-23-05038-f007:**
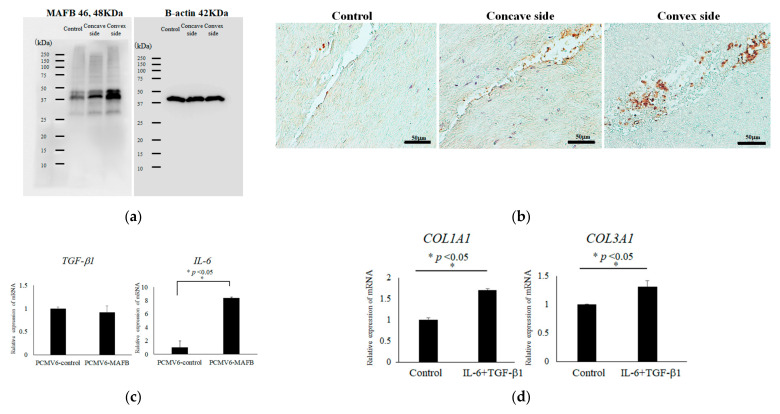
Expression and functional analysis of MAFB. (**a**) Western blotting; (**b**) Immunohistochemical analysis of control (left), concave (center), and convex (right) sides; (**c**) Expression analysis of TGF-beta1 and IL-6 in the presence of elevated MAFB in LF cells. (**d**) Expression analysis of collagens in the presence of additional IL-6 (1 ng/mL) and TGF-beta1 (1 ng/mL) in control LF cells.

**Figure 8 ijms-23-05038-f008:**
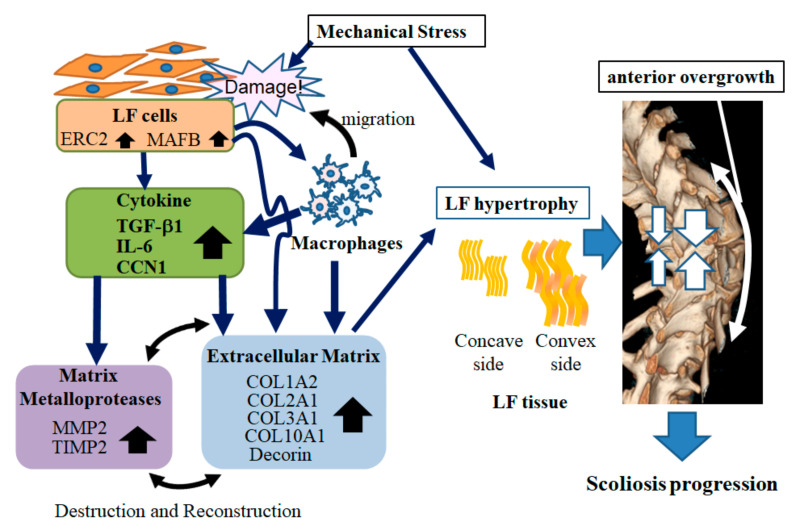
Scheme of progression mechanisms in scoliotic curvature.

**Table 1 ijms-23-05038-t001:** Patient’s demographic data.

Age	Sex	Height (cm)	Weight (kg)	BMI	Risser’s Grade	Lenke Type	Degree of Scoliosis (Cobb)	Degree of Rotation (Apical Vertebral Rotation)	Range of Scoliosis	Apex of Main Thoracic Curvature	Level of Measurement
13	F	169.6	49.5	17.2	4	1	55	16	T6-L2	T9/10	T8/9-T10/11
18	F	149	40	18.0	4	2	53	21	T5-T11	T7/8	T6/7-T8/9
14	M	148	47	21.5	1	1	45	8	T6-T10	T8/9	T7/8-T9/10
13	F	151	43	18.9	2	1	54	10	T4-T10	T7/8	T6/7-T8/9
14	F	157	39	15.8	3	1	60	16	T4-T11	T7/8	T6/7-T8/9
14	F	158.3	37.8	15.1	1	2	71	27	T5-L2	T7/8	T6/7-T8/9
17	F	152.4	45.7	19.7	5	2	55	18	T5-T12	T8/9	T7/8-T9/10
12	F	163.1	47.7	17.9	3	2	60	20	T5-T11	T7/8	T6/7-T8/9
14	F	157.5	42.4	17.1	2	1	58	15	T4-T11	T8/9	T7/8-T9/10
12	F	149.8	38.5	17.2	2	2	64	18	T4-T11	T8/9	T7/8-T9/10
17	F	160	54.8	21.4	4	1	48	14	T5-L2	T9/10	T8/9-T10/11
14	F	155	46	19.1	4	1	53	16	T6-T11	T7/8	T6/7-T8/9
14	F	155.8	46.4	19.1	4	1	45	17	T4-T10	T6/7	T5/6-T7/8
14	F	158.7	55.7	22.1	4	2	45	15	T5-T11	T7/8	T6/7-T8/9
13	F	154	43	18.1	4	1	47	13	T5-T11	T7/8	T6/7-T8/9
16	M	163.5	43.1	16.1	4	1	52	16	T6-L2	T9/10	T8/9-T10/11
12	F	158.3	37.8	15.1	4	2	49	10	T6-T12	T8/9	T7/8-T9/10
14	F	149	40	18.0	2	2	52	12	T5-T12	T7/8	T6/7-T8/9
15	F	157.4	53.5	21.6	4	1	48	12	T5-T12	T8/9	T7/8-T9/10
14	F	154	43	18.1	4	1	51	14	T6-T12	T9/10	T8/9-T10/11
15	F	156	55	22.6	4	1	54	15	T6-T11	T9/10	T8/9-T10/11
13	F	161	61	23.5	4	1	48	13	T4-L3	T10/11	T9/10-T11/12

## Data Availability

The data presented in this study are available on request from the corresponding author.

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
