# Peer review of "Association of Ligamentum Flavum Hypertrophy with Adolescent Idiopathic Scoliosis Progression—Comparative Microarray Gene Expression Analysis"

_ijms, 2022, doi:10.3390/ijms23095038_

Round 1

Reviewer 1 Report

The manuscript by Seki et al., characterizes the structure and gene expression of the ligamentum flavum with the goal of finding a link between LF hypertrophy and AIS. The idea is novel, but it needs some additional information and results. 

Methods:

1. Please describe demographic details of the 20 patients used as control, and add that information under "Subjects". 

2. Please describe the method used to quantify the ratio of collagen fibers to elastic fibers.

3. The qPCR description states that the relative quantification was done comparing to GAPDH, however, it appears it was done relative to control. Did you use 2^-ddCt? please clarify

Results:

  1. The rationale of the genes analyzed by qPCR in Figure 3 is not clear. Consider adding background information either in introduction, or a rationale in results.
  2. Please revise section 2.3 as results are not clearly explained. What does it mean "the convex site was significantly increased..."?
  3. In section 2.4, authors claim to have tested several proteins using western blot, however, only ERC2 is shown. Can you add the other proteins tested as supplementary information or in the same figure? It is not clear why ERC2 was chosen as the only one among the other targets. 
  4. Figure 5, did you test whether the expression of IL6 was upregulated by overexpression of ERC2? This could be important as the manuscript seems to focus in both IL6 and TGFb1 later on.
  5. If MAFB does not increase the expression of TGFb1, what is the rationale to add both TGFb1 and IL6 to LF cells? What is the individual effect? The conclusion that both are required is not supported by the data presented in Figure 7.

Discussion

Please clarify lines 241-242. Did IL6 increase the expression of MAFB? Those results are not shown.

In your analysis, it is not clear which factor comes first after excessive stretching of the LF.

Are there any reports applying mechanical stimulation to LF cells (tension) and analyzing expression of MAFB or ERC2?

Author Response

Point-by-Point Responses

Comments and Suggestions for Authors

The manuscript by Seki et al., characterizes the structure and gene expression of the ligamentum flavum with the goal of finding a link between LF hypertrophy and AIS. The idea is novel, but it needs some additional information and results.

Methods:

Please describe demographic details of the 20 patients used as control, and add that information under "Subjects".

Response: Thank you for your comment. As you suggested, we added demographic detail of the control patients in Methods section. The two patients of spinal tumor were removed because it wasn't appropriate, as pointed out by another reviewer.

We added the information of control patients in Methods section, lines 91-95 as follows.

“Eighteen adolescent patients who had no scoliosis and had taken a thoracic spine MRI imaged for another reason were selected as controls (12 females, 6 males; mean patient age 14.8). The reasons for taking thoracic MRI were 6 chronic back pain, 2 acute back pain, 3 scapular pain, 3 leg numbness, a leg motor weakness, an atrophy of leg muscle, a chronic hypochondrium pain and a steppage gait.”

Please describe the method used to quantify the ratio of collagen fibers to elastic fibers.

Response: Thank you for your comment. As you suggested, we performed the method used to quantify the ratio of collagen fibers to elastic fibers as previously described (29). We added Methods section, lines 171-179 as follows.

“We performed the method used to quantify the ratio of collagen fibers to elastic fibers as previously described (29). To calculate the area of collagen and elastic fibers after EVG staining, we used the ImageJ software program (National Institutes of Health, Bethesda, MD, USA). As shown in the Supplementary Figure 1, in the case of elastic fiber, after converting EVG staining to 8-bit gray scale, the area of the gray-stained part was calculated using ImageJ. In the case of collagen fiber, the area of the grayscale image was calculated by Image J after black-and-white inversion. Three EVG stains with a cross section perpendicular to the elastic fiber were selected and measured three times to calculate the average value.”  

Supplementary Figure 1

Reference

  1. Saito, T.; Yokota, K.; Kobayakawa, K.; Hara, M.; Kubota, K.; Harimaya, K.; Kawaguchi, K.; Hayashida, M.; Matsumoto, Y.; Doi, T.; Shiba, K.; Nakashima, Y.; Okada, S. Experimental Mouse Model of Lumbar Ligamentum Flavum Hypertrophy. PLoS one. 2017, 12, e0169717. doi: 10.1371/journal.pone.0169717.

The qPCR description states that the relative quantification was done comparing to GAPDH, however, it appears it was done relative to control. Did you use 2^-ddCt? please clarify

Response: Thank you for your comment. As you pointed out, we used ∆∆Ct method. We added the methods section, lines 151-153 as follows.

“Data were normalized to those of GAPDH mRNA, and relative gene expression of LF tissues and cells was determined using ∆∆Ct method.”

Results:

The rationale of the genes analyzed by qPCR in Figure 3 is not clear. Consider adding background information either in introduction, or a rationale in results.

Response: Thank you for your comment. As you suggested, we added the background information in Results sections, lines 288-292 as follows.

“Several types of collagen (type I, II, III, and X), cytokines such as transforming growth factor (TGF)-beta1, Interleukin (IL)-1, IL-6, tissue necrosis factor (TNF)-alfa, matrix metalloproteases (MMPs) such as MMP2, and growth factors such as connective tissue growth factor (CTGF) and platelet derived growth factor (PDGF) have been reported as factors involved in LF hypertrophy [28, 29].”

Please revise section 2.3 as results are not clearly explained. What does it mean "the convex site was significantly increased..."?

Response: Thank you for your comment. As you suggested, we revised the sentences in section 3.3, lines 297-298 as follows. “The convex side of ECM protein was significantly increased compared to the concave side (p < 0.05).”

In section 2.4, authors claim to have tested several proteins using western blot, however, only ERC2 is shown. Can you add the other proteins tested as supplementary information or in the same figure?

Response: Thank you for your comment. Although we have tried western blots for some proteins, it was considered that it is better not to publish because it is not an image that can prove a sufficient difference. It was considered meaningless to discuss these images.

It is not clear why ERC2 was chosen as the only one among the other targets.

Response: Thank you for your comment. ERC2, which is one of the other candidates, was analyzed for its function because the expression level was clearly different between the two groups by western blotting and it was also proved in the protein. It was determined to be more significant than other targets because functional analysis clearly demonstrated its role.

Figure 5, did you test whether the expression of IL6 was upregulated by overexpression of ERC2? This could be important as the manuscript seems to focus in both IL6 and TGFb1 later on.

Response: Thank you for your comment. As you suggested, it is an important point as our manuscript whether the expression of IL6 was upregulated by overexpression of ERC2. Therefore, we investigated whether the overexpression of ERC2 increased IL6 in LF cells. We found that it significantly increased the expression of IL6 as shown in below. We have confirmed this experiment twice and obtained reproducibility.

Therefore, we added this figure and the comments in Results section, lines 393-395 as follows.

“COL1A2, COL2A1, COL3A1, IL-6 and TGF-beta1 were significantly increased in the pres-ence of elevated ERC2 in LF cells. (Figure 5c). Overexpression of ERC2, as well as of COL1A2, COL2A1, COL3A1, IL-6 and TGF-beta1 resulted (Figure 5c), suggesting that overexpression of ERC2 may lead to LF hypertrophy on the convex side of the curvature.”

If MAFB does not increase the expression of TGFb1, what is the rationale to add both TGFb1 and IL6 to LF cells? What is the individual effect? The conclusion that both are required is not supported by the data presented in Figure 7.

Response: Thank you for your comment. Our data showed that MAFB did not significantly increase the expression of TGF-beta1. However, expression analysis in LF tissue showed a significant increase in TGF-beta1 and IL-6 (Figure3). Furthermore, overexpressed ERC2 showed significant increase in TGF-beta1 and IL-6 (Figure5). Although you have mentioned that the necessity of both factors is not supported by the data in Figure 7, what we most wanted to know was the identification of the factors involved in the progression of AIS. Although many factors such as cytokines and genes were identified by expression analysis of LF tissue, we found that the combination of TGF-beta and IL-6 caused an increase in the expression of collagen. Further studies may be needed to investigate the effects of individual factors on LF cells in the future.

Discussion

Please clarify lines 241-242. Did IL6 increase the expression of MAFB? Those results are not shown.

Response: Thank you for your comment. As you pointed out, those results are not shown. The text was incorrect and has been corrected in Discussion section, lines 500-501 as follows.

“Our data showed overexpressed MAFB in LF cells highly induced IL-6.”

In your analysis, it is not clear which factor comes first after excessive stretching of the LF.

Response: Thank you for your comment. After stretching (mechanical stimulation) of the LF, it is considered that ERC2 and MAFB expression may come first because our data suggested that overexpressed MAFB increased the expression of IL-6 and overexpressed ERC2 increased both TGF-b1 and IL-6. Although it is undeniable that it may be a positive feedback loop, these phenomena suggest that ERC2 and MAFB may come first.

Are there any reports applying mechanical stimulation to LF cells (tension) and analyzing expression of MAFB or ERC2?

Response: Thank you for your comment. As far as we have investigated, there have been no reports applying mechanical stimulation to LF cells (tension) and analyzing expression of MAFB or ERC2, and there have been no reports of those factors which involved in LF hypertrophy.

Thank you again.

Reviewer 2 Report

Line 88 –Material and Methods section should be transferred here

Line 102 - Fig 1- LF thickness is presented in mm2 not in mm which is confusing

Fig 1b –measurements on the scan present the area (mm2) not thickness as is described. The methodology of LF thickness measurement is not sufficient

Fig 1c – what do arrows indicate?- if the thickness of the LF why they don’t cover the whole slice?

fig 8 – line 255: The crankshaft phenomenon is a postoperative deformity. In 1973, Dubousset reported the progression of scoliotic deformity after posterior spine fusion in younger patients with paralytic scoliosis. He termed the finding “le phenomene due villebrequin,” translated as “the crankshaft phenomenon,” because it appeared that the spine gradually rotated along the length of the fusion. Using this term in the meaning of progression mechanisms in scoliotic curvature is not correct.

Line 257 – Material and method: please add the table with subjects’ data: age, BMI, degree of scoliosis, range of scoliosis (e.g. T4-T12), degree of rotation, the level of measurement, etc.

Line 272 -  A tumor of the thoracic spinal cord can affect surrounding tissues, including the thickness of the LF, and therefore the inclusion of such patients in a control group is not appropriate

Line 363 – the statement “ AIS likely progresses inducing the overgrowth of the anterior vertebral body by keeping tethering posterior spinal elements due to LF hypertrophy.” It is speculation that can be transferred into the discussion section – it is not the conclusion supported by collected data. There were no measurements of vertebral bodies performed. Furthermore “The LF of the convex side was more hypertrophic than the LF of the concave side.” (line 93) It indicates asymmetry of hypertrophy and maybe cause an asymmetric overgrowth of the anterior part of a vertebral body, or maybe first is the overgrowth of a vertebral body and secondary the hypertrophy of LF.  It wasn’t measured. That is why only one sentence “ERC2  and  MAFB  genes were associated with  LF  hypertrophy through increasing TGF-beta1 and IL-6 in patients with AIS.” should stay as a conclusion.

Author Response

Point-by-Point Responses

Line 88 –Material and Methods section should be transferred here

Response: Thank you for your comment. As you suggested, Material and Methods section were transferred between Introduction and Results section.

Line 102 - Fig 1- LF thickness is presented in mm2 not in mm which is confusing. Fig 1b –measurements on the scan present the area (mm2) not thickness as is described. The methodology of LF thickness measurement is not sufficient

Response: Thank you for your comment. As you suggested, the thickness of LF is sometimes presented in mm, but as shown in Figure 1b, the thickness varies considerably from place to place. Furthermore, the comparative study showed that area of LF was more sensitive measurement than the just thickness of LF (ref 27). Therefore, in order to analyze as comprehensively as possible, we measured the thickening of the LF using the cross-sectional area by MRI as described in ref 27 and 28. Therefore, LF thickness of our data presented in mm2.

We added the above to the Method section, lines 108-111 as follows. The cross-sectional area of LF was more sensitive measurement than the thickness of LF. In order to analyze as comprehensively as possible, we measured the thickening of the LF using the cross-sectional area by MRI.

Reference

27.Kim, Y.U.; Park, J.Y.; Kim, D.H.; Karm, M.H.; Lee, J.Y.; Yoo, J.I.; Chon, S.W.; Suh, J.H. The Role of the Ligamentum Flavum Area as a Morphological Parameter of Lumbar Central Spinal Stenosis. Pain Physician. 2017, 20, E419-E424.

28.Takeda, H.; Nagai, S.; Ikeda, D.; Kaneko, S.; Tsuji, T.; Fujita, N. Collagen profiling of ligamentum flavum in patients with lumbar spinal canal stenosis. J. Orthop Sci. 2021, 26, 560-565. doi: 10.1016/j.jos.2020.06.006.

Fig 1c – what do arrows indicate?- if the thickness of the LF why they don’t cover the whole slice?  

Response: Thank you for your comment. This figure shows coronal view of LF after a surgically resection. Arrows indicates bone attachment sites (enthesis). As you pointed out, the axial view of MRI does not represent whole slice of the LF. Therefore, we also measured and confirmed the intervertebral space above and below the apical region. These data also reconfirmed the LF hypertrophy of the convex side.

fig 8 – line 255: The crankshaft phenomenon is a postoperative deformity. In 1973, Dubousset reported the progression of scoliotic deformity after posterior spine fusion in younger patients with paralytic scoliosis. He termed the finding “le phenomene due villebrequin,” translated as “the crankshaft phenomenon,” because it appeared that the spine gradually rotated along the length of the fusion. Using this term in the meaning of progression mechanisms in scoliotic curvature is not correct.

Response: Thank you for your comment. As you suggested, the term “crankshaft phenomenon” may not be correct here. Therefore, we deleted it as shown in Figure 8.

Line 257 – Material and method: please add the table with subjects’ data: age, BMI, degree of scoliosis, range of scoliosis (e.g. T4-T12), degree of rotation, the level of measurement, etc.

Response: Thank you for your comment. As you suggested, we added the demographic data of scoliosis patients about age, BMI, degree of scoliosis, range of scoliosis, degree of rotation, and  the level of measurement in Table 1 as follows.

Table 1. Patient’s demographic data.

Age

Sex

Height

(cm)

Weight

(kg)

BMI

Risser's grade

Lenke type

Degree of scoliosis (Cobb)

Degree of rotation (apical vertebral rotation)

Range of scoliosis

Apex of main thoracic curvature

Level of measurement

13

F

169.6

49.5

17.2

4

1

55

16

T6-L2

T9/10

T8/9-T10/11

18

F

149

40

18.0

4

2

53

21

T5-T11

T7/8

T6/7-T8/9

14

M

148

47

21.5

1

1

45

8

T6-T10

T8/9

T7/8-T9/10

13

F

151

43

18.9

2

1

54

10

T4-T10

T7/8

T6/7-T8/9

14

F

157

39

15.8

3

1

60

16

T4-T11

T7/8

T6/7-T8/9

14

F

158.3

37.8

15.1

1

2

71

27

T5-L2

T7/8

T6/7-T8/9

17

F

152.4

45.7

19.7

5

2

55

18

T5-T12

T8/9

T7/8-T9/10

12

F

163.1

47.7

17.9

3

2

60

20

T5-T11

T7/8

T6/7-T8/9

14

F

157.5

42.4

17.1

2

1

58

15

T4-T11

T8/9

T7/8-T9/10

12

F

149.8

38.5

17.2

2

2

64

18

T4-T11

T8/9

T7/8-T9/10

17

F

160

54.8

21.4

4

1

48

14

T5-L2

T9/10

T8/9-T10/11

14

F

155

46

19.1

4

1

53

16

T6-T11

T7/8

T6/7-T8/9

14

F

155.8

46.4

19.1

4

1

45

17

T4-T10

T6/7

T5/6-T7/8

14

F

158.7

55.7

22.1

4

2

45

15

T5-T11

T7/8

T6/7-T8/9

13

F

154

43

18.1

4

1

47

13

T5-T11

T7/8

T6/7-T8/9

16

M

163.5

43.1

16.1

4

1

52

16

T6-L2

T9/10

T8/9-T10/11

12

F

158.3

37.8

15.1

4

2

49

10

T6-T12

T8/9

T7/8-T9/10

14

F

149

40

18.0

2

2

52

12

T5-T12

T7/8

T6/7-T8/9

15

F

157.4

53.5

21.6

4

1

48

12

T5-T12

T8/9

T7/8-T9/10

14

F

154

43

18.1

4

1

51

14

T6-T12

T9/10

T8/9-T10/11

15

F

156

55

22.6

4

1

54

15

T6-T11

T9/10

T8/9-T10/11

13

F

161

61

23.5

4

1

48

13

T4-L3

T10/11

T9/10-T11/12

Line 272 -  A tumor of the thoracic spinal cord can affect surrounding tissues, including the thickness of the LF, and therefore the inclusion of such patients in a control group is not appropriate

Response: Thank you for your comment. As you suggested, a tumor of the thoracic spinal cord may affect surrounding tissues. We deleted the two subjects of a spinal tumor, and reanalyzed it as shown in Figure 1d.

Line 363 – the statement “ AIS likely progresses inducing the overgrowth of the anterior vertebral body by keeping tethering posterior spinal elements due to LF hypertrophy.” It is speculation that can be transferred into the discussion section – it is not the conclusion supported by collected data. There were no measurements of vertebral bodies performed.

Furthermore “The LF of the convex side was more hypertrophic than the LF of the concave side.” (line 93) It indicates asymmetry of hypertrophy and maybe cause an asymmetric overgrowth of the anterior part of a vertebral body, or maybe first is the overgrowth of a vertebral body and secondary the hypertrophy of LF.  It wasn’t measured. That is why only one sentence “ERC2  and  MAFB  genes were associated with  LF  hypertrophy through increasing TGF-beta1 and IL-6 in patients with AIS.” should stay as a conclusion.

Response: Thank you for your comment. “Asymmetrical of LF hypertrophy maybe cause an asymmetric overgrowth of the anterior part of a vertebral body” is the very interesting points. Basically, the LF hypertrophy in posterior elements is larger than control, and is considered to be part of the anterior overgrowth, but as you mentioned, it is speculation. Therefore, as you suggested, only one sentence, “ERC2 and MAFB genes were associated with LF hypertrophy through increasing TGF-beta1 and IL-6 in patients with AIS” was stayed as a conclusion. We transferred the sentence “AIS likely progresses inducing the overgrowth of the anterior vertebral body by keeping tethering posterior spinal elements due to LF hypertrophy.” into discussion section, lines 512-513.

Thank you again.

Round 2

Reviewer 2 Report

Dear Authors, thank you for your corrections. I recommend the manuscript to publish in IJMS